DOI: 10.1038/s41467-018-04705-8　　**OPEN**

# Calmodulin shuttling mediates cytonuclear signaling to trigger experience-dependent transcription and memory

Samuel M. Cohen[1,2], Benjamin Suutari[2,3], Xingzhi He[1], Yang Wang[1], Sandrine Sanchez[2], Natasha N. Tirko[2], Nataniel J. Mandelberg[2], Caitlin Mullins[2], Guangjun Zhou[1], Shuqi Wang[1], Ilona Kats[2], Alejandro Salah[2], Richard W. Tsien[2,3] & Huan Ma[1]

Learning and memory depend on neuronal plasticity originating at the synapse and requiring nuclear gene expression to persist. However, how synapse-to-nucleus communication supports long-term plasticity and behavior has remained elusive. Among cytonuclear signaling proteins, γCaMKII stands out in its ability to rapidly shuttle $Ca^{2+}$/CaM to the nucleus and thus activate CREB-dependent transcription. Here we show that elimination of γCaMKII prevents activity-dependent expression of key genes (*BDNF*, *c-Fos*, *Arc*), inhibits persistent synaptic strengthening, and impairs spatial memory in vivo. Deletion of γCaMKII in adult excitatory neurons exerts similar effects. A point mutation in γCaMKII, previously uncovered in a case of intellectual disability, selectively disrupts CaM sequestration and CaM shuttling. Remarkably, this mutation is sufficient to disrupt gene expression and spatial learning in vivo. Thus, this specific form of cytonuclear signaling plays a key role in learning and memory and contributes to neuropsychiatric disease.

[1] Institute of Neuroscience, and Department of Neurology of Second Affiliated Hospital, Mental Health Center, NHC and CAMS Key Laboratory of Medical Neurobiology, Zhejiang University School of Medicine, Hangzhou 310058, China. [2] NYU Neuroscience Institute and Department of Neuroscience and Physiology, NYU Langone Medical Center, New York, NY 10016, USA. [3] Center for Neural Science, New York University, New York, NY 10003, USA. These authors contributed equally: Samuel M. Cohen, Benjamin Suutari, Xingzhi He. Correspondence and requests for materials should be addressed to H.M. (email: mah@zju.edu.cn) or to R.W.T. (email: richard.tsien@nyumc.org)

Long-term plasticity and learning requires activity-dependent nuclear transcription[1,2], a form of regulated gene expression that contributes to a critical dialog between synapse and nucleus[3–5]. Excitation-induced recruitment of nuclear transcription factors is exemplified by the phosphorylation of calcium- and cAMP response element binding protein (CREB), a transcription factor particularly important for learning and memory[6,7]. Genetic manipulation of nuclear CREB by mutation of Ser133 to phosphomimetic or non-phosphorylatable forms dramatically alters synaptic plasticity and spatial memory[8]. This focuses attention on the line of communication initiated by the opening of post-synaptic ligand- and voltage-gated channels[7] and culminates in the phosphorylation and resulting activation of CREB[8,9] and subsequent expression of plasticity-related genes[10]. Multiple hypotheses concerning the molecular basis of this communication have been advanced[11–14], some invoking nucleus-directed translocation of signaling molecules like NFκB, Rsk2, Jacob, and CRTC1[14–19]. However, manipulation of these molecules produces confounding changes in brain development and anatomy[20–23], basal synaptic properties[21], or excitability[17]. Thus, clear mechanistic links between neuronal activity, nuclear CREB phosphorylation, and memory consolidation have not been established, leading to consideration of the idea that no translocating protein is required at all[11–14].

To address questions surrounding the relevance of nuclear signaling for synaptic plasticity, gene expression and behavior, while avoiding ambiguous effects from the elimination or sequestration of multifunctional signaling molecules, we sought to manipulate the transport of the signaling protein while sparing the protein itself. We applied this strategy to the nuclear translocation of Ca$^{2+}$/calmodulin (CaM)[18,19,24], which can switch on a nuclear-resident CaM kinase cascade, thus activating both CREB and CREB binding protein (CBP). Synaptic plasticity and spatial memory are both drastically affected by sequestration of nuclear Ca$^{2+}$/CaM[24] or by inhibition of CaMKIV[25,26], the CREB kinase activated by Ca$^{2+}$/CaM. To interfere with the Ca$^{2+}$/CaM-dependent path of communication between synapse and nucleus in vivo, we manipulated CaMKII gamma (γCaMKII), the Ca$^{2+}$/CaM shuttle protein in excitatory neurons[19,27].

We show that genetic deletion of γCaMKII severely impairs activity-dependent expression of key plasticity genes (BDNF, c-Fos, Arc), persistent synaptic strengthening (late LTP or L-LTP), and memory, without detectable changes in anatomy. The necessity of Ca$^{2+}$/CaM translocation per se was further established by introducing a γCaMKII point mutation discovered in an intellectually disabled boy[28]. The mutant γCaMKII (R292P) trapped Ca$^{2+}$/CaM for only seconds rather than minutes, reached the nucleus without its Ca$^{2+}$/CaM cargo, and failed to support gene expression, L-LTP or long-term memory. Thus, signaling to the nucleus by Ca$^{2+}$/CaM translocation supports learning and memory and shows vulnerability in neuropsychiatric disease.

## Results

**γCaMKII is critical for gene expression and L-LTP and learning.** γCaMKII knockout (KO) mice were fully viable[29] and displayed no detectable developmental, morphological or histological brain abnormalities (Supplementary Fig. 1a, b; see also Supplementary Fig. 3, below), minimizing concern about confounding effects on brain development. A custom antibody raised against γCaMKII confirmed the absence of γCaMKII in the γCaMKII KO brain (Supplementary Fig. 1c). Levels of α−, β−, and δCaMKII, as well as other critical neuronal activity-related proteins, were no different than in wild-type (WT) littermates (Supplementary Fig. 1d, e). Neuronal morphology, assessed by Golgi staining of pyramidal neuron dendrites, was not detectably

altered (Supplementary Fig. 1f). To assess the role of γCaMKII in the brain, we subjected γCaMKII KO mice to an extensive battery of behavioral tests (Supplementary Fig. 2). γCaMKII KO mice displayed pronounced impairments in multiple spatial learning tasks designed to assess long-term memory, including the Morris Water Maze (MWM)[3,8,24,30,31], the radial arm maze (RAM), and the inhibitory avoidance task, each of which invoke different motivations and motor systems. In MWM testing, γCaMKII KO mice were deficient in the learning of platform location compared to WT littermates; latency to escape improved more slowly in KO animals and remained longer than WT even after 3–4 days of training (Fig. 1a). In a probe trial (hidden platform removed) on day 5, γCaMKII KO mice took ~5-fold longer to reach the former platform zone and entered that zone half as often (Fig. 1a), suggesting shortcomings in prior memorization.

To look for biochemical underpinnings of the deficits in learning and memory[3,9,32,33], we monitored the experience-driven expression of three major CREB-dependent target genes, BDNF, c-Fos, and Arc (Fig. 1b and Supplementary Fig. 1g). Before training, their expressed protein levels in vivo were similar in WT and γCaMKII KO hippocampus. One hour after MWM training, WT mice displayed a significant increase in hippocampal expression of these genes as compared to naïve animals, assessed by immunoblot (Fig. 1b, black bars), consistent with previous reports[10]. In contrast, deletion of γCaMKII prevented the training-induced elevation of expression of all three genes ($p > 0.4$, Fig. 1b, rightmost red bars). Thus, γCaMKII is critical for experience-dependent expression of CREB target genes over the course of spatial learning in vivo.

A similar comparison was made between behavior and gene expression for an inhibitory avoidance (IA) task, in which animals were mildly shocked following entry into the dark compartment of a 2-compartment chamber. WT and γCaMKII KO mice were indistinguishable in the latency to explore the dark compartment during trials prior to administration of a mild shock (dashed lines, Fig. 1c), and in test sessions if no shock was given in the training sessions (Supplementary Fig. 2a). Upon further testing 24 h later, WT animals reliably avoided the conditioned context whereas γCaMKII KO mice were deficient in this avoidance behavior (Fig. 1c, $p < 0.05$, Kolmogorov–Smirnov test). Correspondingly, the proportion of c-Fos-positive pyramidal neurons was elevated in the WT controls but not in the γCaMKII KO group. Thus, the inhibitory avoidance task provided another example of faulty learning associated with a lack of enduring gene expression. In both hippocampal-dependent tasks (MWM and IA), the absence of γCaMKII engendered clear deficiencies in behavioral performance, comparable to deficits produced by lesions to hippocampal area CA1[34].

To clarify the mechanism underlying the deficit in learning, we next asked whether disruption of γCaMKII signaling affects hippocampal late-LTP (L-LTP). This enduring form of synaptic plasticity depends on transcription and translation and supports spatial memory formation[5,6,24,35,36]. We induced L-LTP at Schaffer collateral-CA1 synapses with a standard protocol, three trains of tetanic stimulation (100 Hz, 1 s) spaced by 5 min[3,24,31,37], in hippocampal slices derived from WT and γCaMKII KO mice. While no deficit was observed in early LTP (E-LTP) between genotypes, L-LTP was strongly affected (Fig. 1d): the initial potentiation in γCaMKII KO slices decayed much faster than that in WT, with a clear difference emerging at 2 h post tetanization (Fig. 1d) (WT: 140.7 ± 5.5%, KO: 121.5 ± 3.9%, $n = 6$ mice, $p = 0.02$), and growing up to the end of recording (WT: 134.9 ± 5.9%, KO: 105.9 ± 8.4%, $n = 6$ mice, $p = 0.02$). In contrast, no differences were found in basal properties of WT and γCaMKII KO slices (Fig. 1d): synaptic input–output relationship (ratios of fEPSP slope to fiber volley of 4.61 ± 0.81 for WT vs.

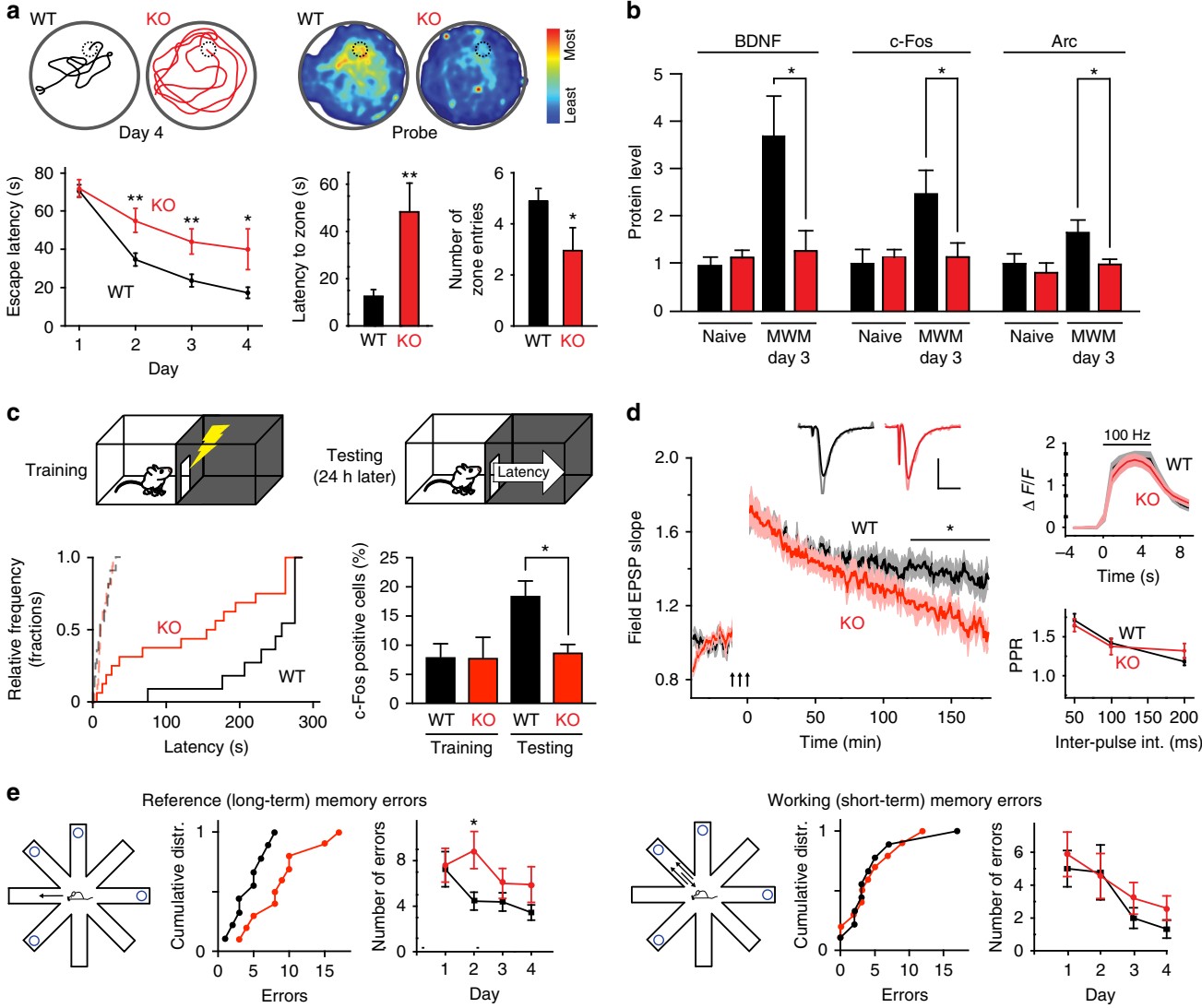

**Fig. 1** Spatial memory, CREB-dependent gene expression, L-LTP are impaired in γCaMKII-deficient mice. **a** Spatial memory acquisition and retrieval were tested with the Morris Water Maze (MWM). Bottom left, mean escape latencies to reach a hidden platform were plotted against training day (4 trials per day) for WT mice (black, $n = 21$ for 1st 3 days, 12 for 4th day) and γCaMKII KO mice (red, $n = 19$ for 1st 3 days, 10 for 4th day). Top left, representative path used by mice to reach hidden platform (dotted circle) during the last trial of Day 4. Top right, activity histogram representing average proportion of total time spent by mice during probe trial; γCaMKII KO mice ($n = 10$), WT ($n = 12$); scale at far right, amount of time spent at different locations. Bottom middle, latency to reach platform zone during probe trial. Bottom right, number of zone entries during probe trial. **b** Western blot analysis of BDNF, c-Fos, and Arc expression in the hippocampus of WT (black, $n = 9$) and KO (red, $n = 7$) mice. Animals sacrificed 1 h after end of MWM training on 3rd day. **c** Inset, schematic depiction of inhibitory avoidance test. Left graph, cumulative distribution of latencies to enter the dark compartment before conditioning (dashed lines) and 24 h post conditioning (solid lines). Conditioning done by administration of a mild shock in the dark compartment. Right bar graph, percentage of c-Fos+ neurons in CA1 assessed 1 h after test trial. **d** Left, late-phase LTP (L-LTP) induced by three trains of tetanic stimulation (100 Hz, 1 s), spaced by 5 min intervals[24,37], with timing denoted by the gap in the field EPSP trace ($n = 6$ mice for each group). Superimposed representative EPSPs show the basal EPSP (bold color) and the EPSP at 165 min (muted color) after L-LTP induction. Calibration bars, 1 mV and 10 ms. Top right, Fluo-4 $Ca^{2+}$ responses ($\Delta F/F$) to 5 s of 100 Hz field stimulation in acute hippocampal slices from wild-type and γCaMKII KO mice. Bottom right, paired-pulse ratio (slope of second EPSP/slope of first EPSP) plotted with interpulse intervals of 50, 100, and 200 ms ($n \geq 9$ slices). **e** Radial arm maze behavior. Reference and working memory on day 2, reflecting various aspects of spatial memory 24 h after first exposure to maze, were assayed respectively by recording the number of entries into arms that were never baited (reference memory) or re-entries into arms that were initially baited (working memory) ($n \geq 9$ for both genotypes). Statistical analysis was performed with one-way ANOVA followed by Holm–Sidak post hoc test unless otherwise noted. *$p \leq 0.05$, **$p \leq 0.01$, ***$p \leq 0.001$. Error bars represent SEM

4.58 ± 0.57 for γCaMKII KO, $p > 0.9$), paired-pulse ratio, or somatic $Ca^{2+}$ influx in response to 100 Hz stimulation[38].

Our finding of reduced L-LTP with intact E-LTP suggested that γCaMKII KO mice might display deficits in long-term (reference) memory but not short-term (working) memory. In the radial arm maze, γCaMKII KO mice were no different than WT animals upon first presentation of arm locations (reference "errors",

day 1), but failed to show memory savings 24 h later (Fig. 1e). In contrast, KO and WT mice committed similar numbers of short-term (working memory) errors. Likewise, further tests in the MWM indicated that deficiency in γCaMKII KO performance specifically reflected a spatial learning deficit; our behavioral results could not be otherwise explained by reluctance to swim, swimming speed, wall-hugging behavior, muscle function

(forelimb-hanging test), locomotor activity (open field test), or anxiety (tendency to thigmotaxis, open field test and elevated zero maze) (Supplementary Fig. 2). Taken together, these results suggest that γCaMKII plays a critical role in a series of events encompassing experience-dependent gene expression, L-LTP and long-term memory in multiple settings.

**γCaMKII downstream of $Ca_V1$/NMDARs regulates spatial learning.** To narrow down the requirements for γCaMKII signaling, we selectively deleted γCaMKII in excitatory neurons late in development, using offspring of γCaMKII$^{LoxP/LoxP}$ mice crossed with αCaMKII-CRE mice (exc-KO, Fig. 2a). Hippocampal CA1 pyramidal neurons so deprived of γCaMKII were no different than WT with regard to passive membrane properties, excitability, dendritic branching (Supplementary Fig. 3c) or cellnumber (Supplementary Fig. 4a). Nonetheless, selective

elimination of γCaMKII in excitatory neurons impaired spatial learning (Fig. 2a), recapitulating the deficiency in MWM performance observed with global γCaMKII deletion ($p = 0.5$, Fig. 1a). Befitting the role of γCaMKII as a surface-to-nucleus shuttle for $Ca^{2+}$/CaM[19], training-induced CaM translocation, seen as increased nuclear CaM immunostaining in WT mice, was eliminated in γCaMKII exc-KO animals (Fig. 2b, c). Likewise, elevation of nuclear c-Fos, reflective of training-induced gene expression in vivo[10], was similarly inhibited (Fig. 2b, c). In contrast, basal levels of expression of c-Fos, Arc, BDNF, or nuclear CaM were no different in γCaMKII exc-KO mice and WT littermates (Supplementary Fig. 4), indicating that γCaMKII acted mainly under conditions of training.

These results raised questions about signaling by NMDA receptors, whose importance for induction of L-LTP and learning is well known[31,32] but whose linkage to nuclear shuttling of $Ca^{2+}$/CaM by γCaMKII remains mysterious. To address this,

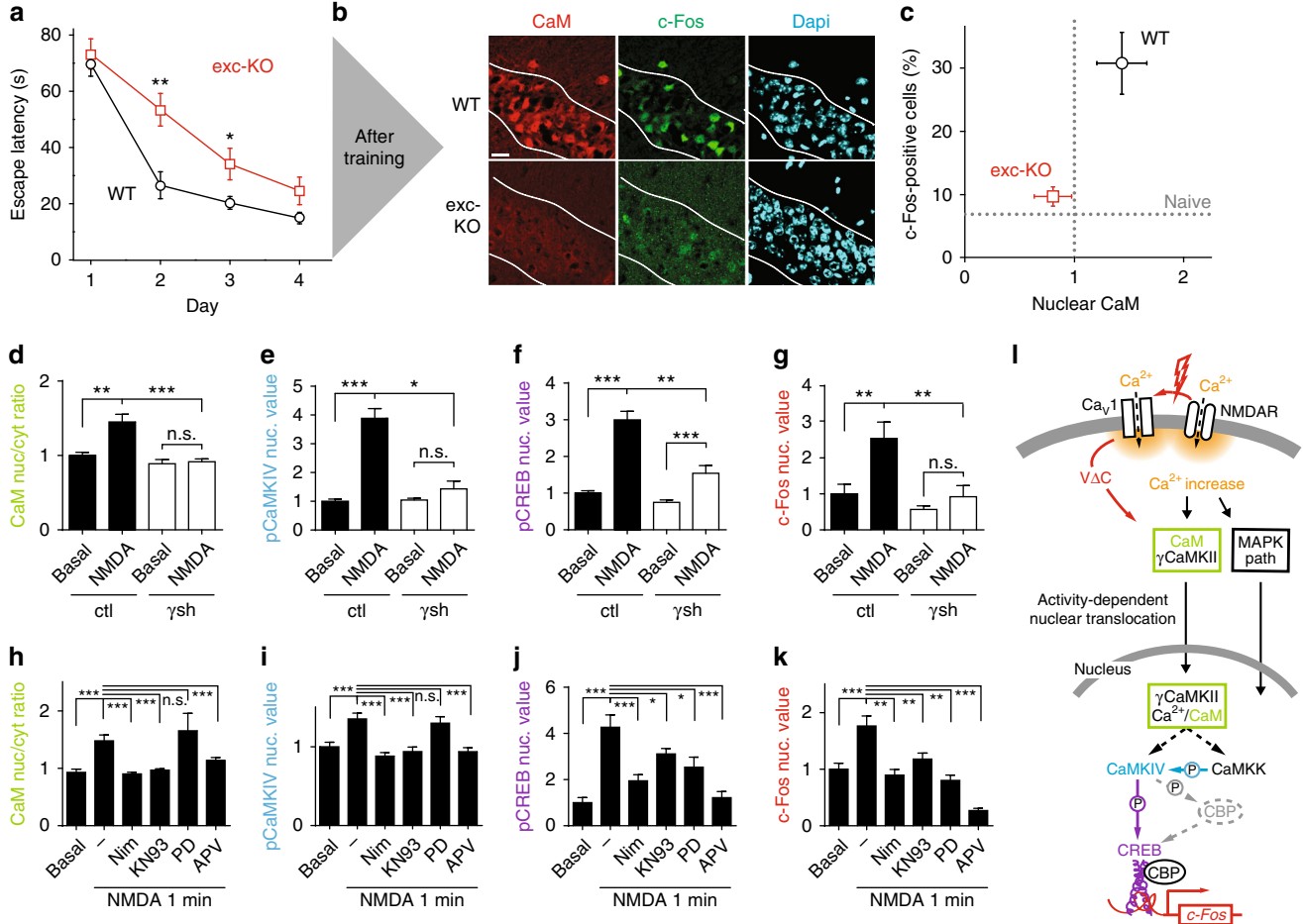

**Fig. 2** Spatial memory and gene expression depend on NMDAR- and $Ca_V1$-mediated signaling via γCaMKII/CaM translocation. **a** Significant impairment in the ability of γCaMKII exc-KO mice during the acquisition phase of the MWM test ($n = 12$ mice for each group). No difference was detected between behavior of exc-KO and KO animals ($p > 0.16$, one-way ANOVA) or between the two cohorts of WT mice ($p > 0.15$, one-way ANOVA). **b** Representative images show nuclear CaM and c-Fos expression in the pyramidal layer of CA1 (denoted by white lines) of WT mice at 1 h after MWM training, which are absent in γCaMKII exc-KO mice. Scale bar, 20 μm. **c** Percentage of c-Fos-positive cells plotted against nuclear CaM level in CA1 of WT or exc-KO mice, 1 h after MWM training ($n = 5$ mice for each group, $p = 0.04$ for c-Fos and $p = 0.006$ for CaM). Vertical and horizontal dotted lines represent CaM and c-Fos nuclear values in naive (untrained) mice. **d–k** Cultured cortical neurons were either "mock-stimulated" (basal) or stimulated with 25 μM NMDA and 5 μM glycine, without (control, ctl) or with γCaMKII shRNA knockdown (**d–g**), or without (control, labeled '-') or with nimodipine, KN93, PD98059, or APV. Following fixation, cells were stained for CaM (**d**, **h**), pCaMKIV (**e**, **i**), pCREB (**f**, **j**), or c-Fos (**g**, **k**). All plots are the average of two experiments with >25 cells per condition (one-way ANOVA followed by Student's t-test). **l** Schematic diagram of $Ca_V1$- and NMDA receptor-driven, γCaMKII-mediated cytoplasm-to-nucleus signaling[19,27,44]. Colors refer to signaling intermediates assessed in **d–k**. For more details on alternative pathways, see Supplementary Fig. 5m. Statistical analysis was performed with one-way ANOVA followed by Holm–Sidak post hoc test unless otherwise noted. *$p \leq 0.05$, **$p \leq 0.01$, ***$p \leq 0.001$. Error bars represent SEM

we exposed cultured hippocampal or cortical neurons to 25 μM NMDA and 5 μM glycine to activate NMDARs, as confirmed with NMDAR-selective blocker APV (Fig. 2d–k and Supplementary Fig. 5a, b). NMDA stimulation elevated nuclear CaM (reflected by the nucleus/cytoplasm ratio, nuc/cyt) and increased pCaMKIV. Involvement of γCaMKII in NMDA-dependent cytonuclear signaling was tested by infecting neurons with viruses encoding a specific γCaMKII shRNA or with a nonsilencing control[19]. Increases in nuclear CaM, pCaMKIV,

pCREB, and c-Fos evoked by NMDA stimulation were prevented or largely reduced by the γCaMKII shRNA (Fig. 2d–g). Thus, the γCaMKII pathway supported NMDA receptors in signaling to the nucleus (Fig. 2l).

Further experiments clarified how signaling by γCaMKII relates to the classical MAP kinase pathway. NMDA-driven elevation of nuclear CaM and CaMKIV activity were prevented by nimodipine (inhibits $Ca_V1$) or KN93 (inhibits CaMKs), but not PD98059 (inhibits MAP2K1/MEK1, enzymes upstream of

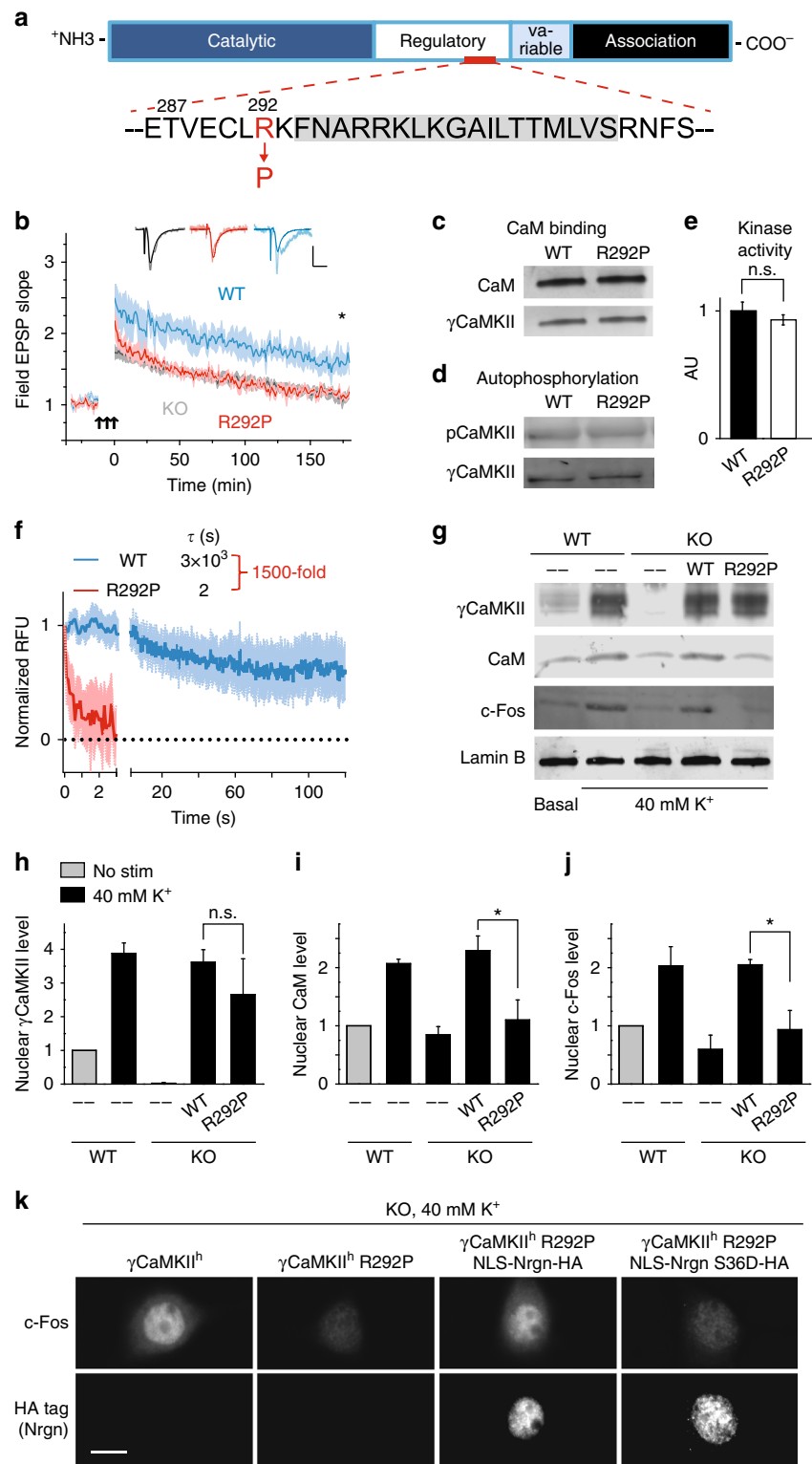

ERK) (Fig. 2h, i). NMDA stimulation also triggered an increase in pCREB that was prevented by nimodipine or KN93 (Fig. 2j), and also reduced by PD98059, consistent with involvement of both CaMK and MAPK pathways[39] (see ref. [39] for earlier references). NMDA-driven elevation of c-Fos protein, assayed 40 min following stimulation to allow time for transcription and translation, was prevented by nimodipine, KN93, or PD98059 (Fig. 2k).

These results indicate that NMDAR activation drives divergent γCaMKII- and MAPK-mediated pathways that reconverge to control gene expression. We verified that deletion of γCaMKII spared critical markers of the MAPK signaling branch (elevation of nuclear levels of pMAPK, Rsk2, CRTC1; Supplementary Fig. 5). Thus, the dramatic behavioral consequences of γCaMKII KO (Figs. 1a, c, e and 2a) are specifically indicative of the role of γCaMKII-mediated branch of the signaling to the nucleus.

**ID-associated γCaMKII R292P mutation impairs CaM shuttling**. To gain further insight into the functional significance of cytonuclear communication by protein translocation, we focused on genetic variation in γCaMKII that alters human intellectual performance[28]. Extending an early study of heritability of intellectual disability (ID) that implicated γCaMKII[40], de Ligt et al.[28] performed exome sequencing of a boy with severe ID and his unaffected parents and discovered a de novo coding mutation in γCaMKII (R292P, Fig. 3a). The positively charged residue at position 292 is highly conserved across CaMKII isoforms (Supplementary Fig. 6a) and neighbors a largely α-helical region required for CaM binding[41] (Fig. 3a); mutation to proline could disrupt that region and its function. Accordingly, we reasoned that this ID-linked mutation might illuminate the mechanism by which γCaMKII supports learning and memory.

Lentiviral expression of human γCaMKII R292P in hippocampal slices from KO mice failed to rescue L-LTP (red trace, Fig. 3b); early potentiation and late-phase decay were indistinguishable from KO slices (black). In contrast, slices expressing human WT γCaMKII (blue) showed strong early potentiation that was maintained throughout the recording, and was significantly greater than that with γCaMKII R292P. The disabling effect of the R292P mutation on synaptic plasticity prompted us to analyze its impact on biochemical signaling in vitro (Fig. 3c–f) and in neurons (Fig. 3g–j). Both purified WT and R292P γCaMKII readily bound CaM as seen with CaM overlays (Fig. 3c), underwent phosphorylation at T287 as detected by pCaMKII-specific immunoblotting (Fig. 3d), and displayed catalytic activity as a protein kinase (Fig. 3e). Beyond these similarities between WT and mutant molecules, we noted a potentially critical contribution of γCaMKII in which its kinase activity is dispensable but its ability to trap and translocate CaM to the nucleus is crucial[19]. Knowing that phosphorylation of αCaMKII at T286 increased its affinity for $Ca^{2+}$/CaM >1000-fold, largely because of slowed CaM dissociation[42,43], we characterized such "CaM trapping" by WT and R292P mutant forms of γCaMKII. The dissociation of fluorescently labeled CaM (CaM $(C75)_{IAE}$) from T287-phosphorylated γCaMKII was monitored as a decrease in fluorescence upon sudden addition of non-fluorescent CaM[41]. Strikingly, dissociation from the R292P mutant was ~1500-fold faster than from WT γCaMKII, occurring within seconds rather than minutes, even in the presence of $Ca^{2+}$ (Fig. 3f). Thus, by the time the mutant γCaMKII began to accumulate in the nucleus (>30 s)[19], CaM would no longer be bound and would therefore not translocate effectively.

To test this prediction, we introduced WT or R292P forms of γCaMKII into cultured hippocampal and cortical neurons from γCaMKII KO mice. Neurons were depolarized with 40 mM $K^+$ and protein levels in isolated nuclei were analyzed. Stimulation-induced increases in c-Fos, nuclear CaM and nuclear γCaMKII were absent in neurons from γCaMKII KO mice (Fig. 3g–j and Supplementary Fig. 6b), as previously found in cultured rat cortical neurons[19]. Each of these events was rescued by re-introduction of WT human γCaMKII. In contrast, γCaMKII R292P was able to translocate to the nucleus, but unable to support nuclear CaM translocation or c-Fos expression (Fig. 3g–j).

Having demonstrated the necessity of intact γCaMKII and CaM shuttling for gene expression, we proceeded to test for sufficiency of nuclear CaM delivery by directly uncaging CaM in the nucleus. The vehicle was neurogranin, an endogenous buffer for apoCaM that releases CaM once it becomes $Ca^{2+}$ bound, that we HA-tagged and targeted to the nucleus (NLS-Nrgn)[19]. Immunocytochemistry verified that introduction of γCaMKII R292P in cultured γCaMKII KO neurons failed to rescue c-Fos expression, consistent with results from isolated

**Fig. 3** De novo point mutation of *CAMK2G* impairs CaM trapping and shuttling by γCaMKII. **a** Schematic of γCaMKII protein, highlighting the putative CaM-trapping region (aa ~286–315 within the regulatory region). Amino acids shaded in gray directly bind $Ca^{2+}$/CaM[41,58,63]. Arg292 to Pro292 mutation (R292P, red) identified as a candidate basis for intellectual disability via family-based exome-sequencing[28]. **b** L-LTP induction in hippocampal slices from γCaMKII KO mice (black) or exc-KO mice injected with lentivirus expressing γCaMKII (blue) or γCaMKII R292P (red). Note overlap between KO and exc-KO + γCaMKII R292P traces. Potentiation at 180 min significantly greater for exc-KO + γCaMKII (198.9 ± 26.8%), than exc-KO + CaMKII R292P (107.3 ± 5.7%) (*n* = 4 mice for each group). Superimposed representative EPSPs show basal EPSP (bold color) and EPSP at 165 min (muted color) after L-LTP induction. Calibration bars, 1 mV and 10 ms. **c** CaM overlay analysis (Experimental Procedures). Purified HA-tagged γCaMKII or γCaMKII R292P was run on SDS-PAGE gel and membrane exposed to 0.5 μg/ml CaM for 1 h in the presence of 1 mM $CaCl_2$. **d** Western blot analysis of phospho-Thr287 using antibodies against pCaMKII and HA tag, following incubation of purified HA-tagged γCaMKII or γCaMKII R292P with 25 μM ATP for 1 min in presence of 1 mM $CaCl_2$ and 1 μM CaM. **e** In vitro phosphorylation of syntide-2 by purified recombinant HA-tagged γCaMKII or γCaMKII R292P was measured using the CyclEx CaM kinase II activity ELISA kit. **f** Dissociation kinetics of phosphorylated γCaMKII and γCaMKII R292P. Traces show that dissociation of CaM$(C75)_{IAE}$ from complexes with phospho-γCaMKII (blue) is 1500-fold faster (*p* = 0.002) than from complexes with phospho-γCaMKII R292P (red). Curves generated by fitting data to two phase decay ($Y = A1^*exp(-t/\tau_1) + A2^*exp(-t/\tau_2) + Y_0$)[41]. Data normalized so intensity difference between kinase-bound and free $Ca^{2+}$/ CaM$(C75)_{IAE}$ would be 1. **g–j** Western blot probing for nuclear γCaMKII, CaM, and c-Fos in isolated nuclei of cultured hippocampal and cortical neurons from WT or γCaMKII KO mice stimulated with 40 mM KCl for 1 h. Both γCaMKII (WT) and γCaMKII R292P translocated to the nucleus (**g**, **h**); however, only re-introduction of transgenic γCaMKII (WT), not γCaMKII R292P, into γCaMKII KO neurons rescued CaM translocation (**g**, **i**) and c-Fos expression (**g**, **j**). **k** c-Fos response in cultured hippocampal and cortical neurons from γCaMKII KO mice stimulated with 40 mM KCl for 1 h. γCaMKII (WT) but not γCaMKII R292P (no HA tag) could restore c-Fos response. Co-expressing γCaMKII R292P with NLS-Nrgn but not NLS-Nrgn S36D was able to restore the c-Fos response, consistent respectively with an ability or inability to harbor CaM for release upon nuclear $Ca^{2+}$ elevation[19]. Scale bar, 10 μm. Statistical analysis performed with one-way ANOVA followed by Holm–Sidak post hoc test unless otherwise noted. *$p \leq 0.05$, **$p \leq 0.01$, ***$p \leq 0.001$. Error bars represent SEM

nuclei. In contrast, introduction of NLS-Nrgn along with γCaMKII R292P fully restored stimulation-induced c-Fos expression (Fig. 3k and Supplementary Fig. 6c, d). The rescue of c-Fos expression failed when the NLS-Nrgn was modified (S36D) to render it incompetent to bind apoCaM and thus release CaM upon $Ca^{2+}$ influx (Fig. 3k and Supplementary Fig. 6c, d). Thus, the failure of γCaMKII R292P could be bypassed by exogenous replacement of nuclear $Ca^{2+}/CaM$ elevation to circumvent the defective shuttle. Given that kinase activity of γCaMKII is dispensable for signaling to the nucleus[19], the critical events for gene expression, operative in WT but lost in the ID-linked mutant, are CaM trapping and nuclear CaM delivery.

**R292P mutation impairs gene expression and learning in vivo.** Returning full circle to spatial learning, we tested whether the behavioral deficit in excitatory neuron-specific γCaMKII KO mice could be repaired by acute expression of γCaMKII in the mature hippocampus (Fig. 4a). Hippocampi of adult WT or γCaMKII exc-KO mice were targeted with bilateral injections of saline or lentiviruses containing WT or mutant human γCaMKII, resulting in levels of expression of γCaMKII three-fold greater than endogenous (Supplementary Fig. 6e). Strikingly, the deficit in MWM spatial learning seen in γCaMKII exc-KO mice was fully rescued by introduction of γCaMKII. To verify that the behavioral repair was associated with rescue of nuclear signaling, we stained CA1 pyramidal cells[31] for CaM and c-Fos 1 h after

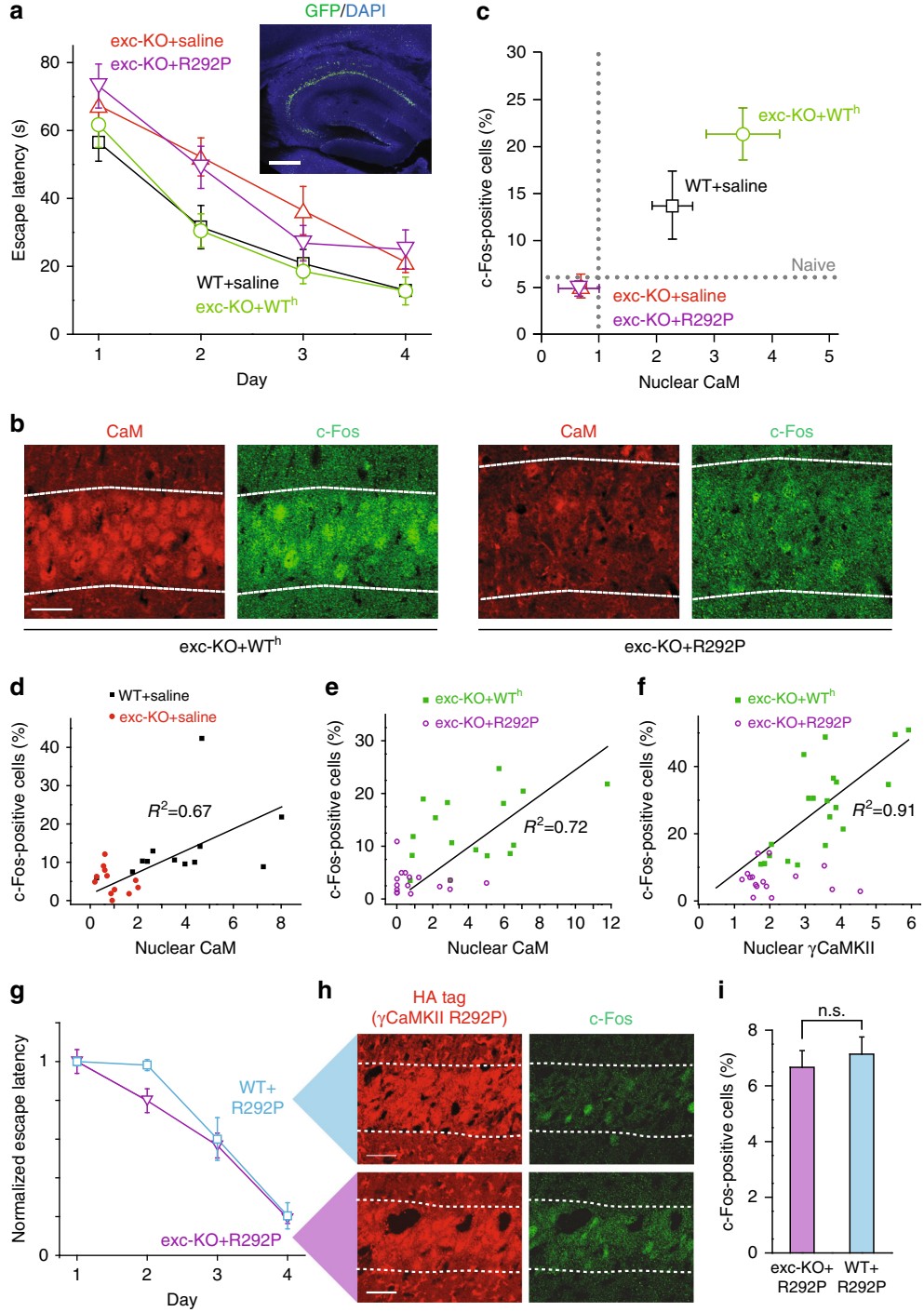

MWM training. Both CaM translocation and c-Fos expression were fully rescued by the re-introduction of γCaMKII (Fig. 4b, c). Hence, behavioral performance depends critically on the availability of γCaMKII during the task, not its presence during brain development.

Unlike WT γCaMKII, γCaMKII R292P failed to rescue the spatial learning deficit when introduced into the mature hippocampus. Mice expressing the mutant performed no differently than exc-KO animals (Fig. 4a). Likewise, they displayed levels of nuclear c-Fos and CaM in CA1 that were no higher than in γCaMKII exc-KO (Fig. 4c; $p > 0.8$) when examined after MWM training. The strong correlation between post-training nuclear CaM and c-Fos in WT mice (Fig. 4d) was rescued in exc-KO mice by expression of human WT γCaMKII (Fig. 4e); likewise, nuclear c-Fos was correlated with γCaMKII (Fig. 4f). In contrast, following introduction of γCaMKII R292P, nuclear c-Fos was not correlated with nuclear CaM (Fig. 4e), nor with nuclear γCaMKII (Fig. 4f), even though basal levels of expressed γCaMKII R292P were not different than expressed γCaMKII (Supplementary Fig. 6e, f). These in vivo results reflect inefficient CaM shuttling by the mutant protein. In all respects, our observations with mutant γCaMKII R292P sharply contrasted with those obtained with γCaMKII.

**γCaMKII R292P demonstrates dominant negative effect**. We next tested for a dominant-negative effect of the R292P mutant, of possible relevance to the pathogenesis of ID in a human heterozygote. Interestingly, overexpressed γCaMKII R292P in cultured WT hippocampal neurons prevented nuclear CaM translocation, suggesting a strong dominant negative effect of γCaMKII R292P in regulating nuclear CaM signaling (Supplementary Fig. 6g). Moreover, following expression of γCaMKII R292P in the hippocampus, escape latencies in the MWM were no faster in WT than in exc-KO animals (Fig. 4g). Experience-dependent c-Fos expression was compared in WT and exc-KO hippocampus following behavioral testing. Unlike the striking difference in experience-dependent c-Fos staining between WT and exc-KO (Fig. 2b), in γCaMKII R292P-infected pyramidal neurons, distinguished by their HA-tag (Fig. 4h), such post-training differences were no longer observable (Fig. 4h, i). Evidently, exogenous expression of mutant γCaMKII can nullify the impact of normal protein with regard to gene expression as well as behavior, consistent with some kind of dominant negative effect.

## Discussion

Here we show the importance of $Ca^{2+}$/CaM translocation to the nucleus for learning and memory, and how this translocation is disrupted by a human mutation linked to intellectual disability. Mediated by γCaMKII in excitatory neurons, the $Ca^{2+}$/CaM shuttle links activation of NMDARs[44] and $Ca_V1$ channels[19] to nuclear transcription, thus supporting L-LTP and long-term memory. Our in vivo experiments indicate that this mechanism is necessary for classic hippocampal CA1-based learning in adult animals. Independence from any possible effects of γCaMKII on early brain development was demonstrated by late deletion of γCaMKII (Fig. 2) and adult rescue by γCaMKII delivery in a knockout background (Fig. 4).

As a mechanism for synaptic activity-dependent, cytonuclear communication based on protein translocation[45–48], $Ca^{2+}$/CaM shuttling can be compared with leading examples of synapto-nuclear signaling (NFκB, RSK2, Jacob, CRTC1)[11–13]. It will be interesting to see if these signaling pathways, well described in neuronal cultures, are also critical for the establishment of memory in vivo, separable from other important but confounding functions distinct from memory storage[17,20–23]. The significance of the γCaMKII/$Ca^{2+}$/CaM shuttle could be cleanly defined via a specific manipulation of shuttle function (Figs. 3 and 4) and extensive control experiments to rule out morphological, biochemical and behavioral side effects (Supplementary Figs. 1–6). This addresses a long-standing challenge to studies of LTP: finding key molecular underpinnings, as functionally critical to neuroplasticity as a door hinge is to an opening door[49], and linking these processes to behavior.

The strong dominant negative effect of γCaMKII R292P (Fig. 4g, h; Supplementary Fig. 6g) is intriguing as it helps to establish a logical connection between our data and the (heterozygous) human patient. A dominant negative effect could result from competition between wild-type and mutant γCaMKII at various critical steps along the shuttling pathway[19]: competition for cross-phosphorylation of γCaMKII by α/βCaMKII, for dephosphorylation by CaN to enable the NLS, or for binding to protein components supporting transport to the nucleus. The dominant negative effect of γCaMKII R292P cannot easily be attributed to loss of catalytic activity because a charge-altering mutation in the homologous position of αCaMKII (K291E) enhances affinity for $Ca^{2+}$/CaM 5-fold, via destabilization of the basal CaM-free state[50]; further, we confirmed that γCaMKII R292P retained kinase activity. Because even kinase-dead γCaMKII is able to support nuclear $Ca^{2+}$/CaM delivery[19], a

---

**Fig. 4** R292P mutation disables the CaM-shuttling function of γCaMKII and inhibits activity-dependent gene expression, late-LTP, and spatial memory formation. **a** MWM test of WT or γCaMKII exc-KO mice 3 weeks after injection with HA-γCaMKII (WT$^h$) lentivirus, HA-γCaMKII R292P lentivirus, or saline bilaterally in the hippocampus ($n = 10$ mice for each group). Inset, representative injection with lentivirus encoding GFP, demonstrating localization of viral infection. Scale bar, 500 μm. **b** CaM and c-Fos staining in CA1 of γCaMKII exc-KO mice injected with HA-γCaMKII or HA-γCaMKII R292P. Slices are from mice used in **a** that were sacrificed 1 h post training on day 4. Scale bar, 20 μm. **c** Percentage of c-Fos-positive cells plotted against mean nuclear CaM in CA1 1 h after MWM training in WT or γCaMKII exc-KO mice injected with saline, or in γCaMKII exc-KO injected with HA-γCaMKII or HA-γCaMKII R292P (analysis after behavior in **a**). Dotted line represents CaM (normalized to naive) or c-Fos nuclear value in naive mice ($n = 3$–4 mice per condition). CaM and c-Fos values were significantly greater ($p = 0.0005$ for CaM and $p < 0.0001$ for c-Fos) in γCaMKII exc-KO mice injected with HA-γCaMKII as compared to those injected with saline or HA-γCaMKII R292P. **d–f** CaM, c-Fos, and γCaMKII (HA-tag) staining in CA1 after MWM training, with each dot representing an individual sliced and stained section of CA1, each containing data from 100–140 pyramidal cells. Nuclear CaM value in **d** and **e** is normalized to exc-KO + saline. **d** Percent of c-Fos-positive cells plotted against mean nuclear CaM in WT (Adj. $R^2 = 0.67$) or γCaMKII exc-KO mice injected with saline 1 h after the end of MWM training. **e** Percent of c-Fos-positive cells is plotted against nuclear CaM in γCaMKII exc-KO mice, injected with either HA-γCaMKII (Adj. $R^2 = 0.72$) or HA-γCaMKII R292P, 1 h after the end of MWM training. **f** Percent of c-Fos-positive cells is plotted against average nuclear γCaMKII (staining against HA tag) in γCaMKII exc-KO mice, injected with either HA-γCaMKII (Adj. $R^2 = 0.91$) or HA-γCaMKII R292P (Adj. $R^2 = -0.053$), 1 h after the end of MWM training. **g–i** γCaMKII exc-KO or WT mice injected with HA-γCaMKII R292P in CA1 were trained in the MWM. **g** No significant differences in escape latency during the acquisition phase of the MWM test. **h** Staining of HA and c-Fos in CA1 in mice sacrificed 1 h post training. Scale bar, 20 μm. **i** Percentage of c-Fos-positive cells ($n = 2$ mice, $n > 1800$ cells for each condition). Error bars represent SEM

mutation-based deficiency in kinase activity of γCaMKII is unlikely to explain how the R292P mutant interferes with E–T coupling to affect long-term memory.

The functional relevance of the γCaMKII/$Ca^{2+}$/CaM shuttle mechanism is highlighted by human genetics. Each of the molecules along the signaling pathway is encoded by a gene implicated in multiple brain diseases, among them schizophrenia, major depressive disorder, bipolar disorder and autism spectrum disorder. In particular, CAMK2G, the gene encoding γCaMKII, contributes to a genetic cluster that influences memory performance and imaged hippocampal activity in humans[40], and has cropped up in genetic studies of ASD[51,52]. In rodents, γCaMKII has been implicated in depression-like behavior[53] and as a regulatory target of the NMDA receptor-dependent microRNA miR-219[54]. While the ID-linked R292P mutation[28] causes the most functionally precise lesion so far, it will likely be joined by other genetic modifications that illuminate neuropsychiatric disorders along with fundamental aspects of memory and cognition.

## Methods

**Data acquisition**. All analyses were performed with the experimenter blinded to the genotype of mice, cultured neurons and acute slices.

**Animals**. γCaMKII-KO and γCaMKII-exc-KO mice were kindly provided by Dr. Eric N. Olson (UT Southwestern Medical Center) with help of Dr. Johannes Backs and his staff (University of Heidelberg). αCaMKII-Cre mice (Stock No. 005359) were purchased from the Jackson Laboratory and maintained on a C57BL/6J background. The mice (10–14 weeks of age, both genders) were used for behavioral tests. Subject mice were kept in a 20 °C room on a 12-h light/dark cycle (lights off at 6:00 AM) with food and water available ad libitum. Mice were singly housed for at least 7 d before experiments. All experiments were performed during the dark cycle, after acclimatization to the experimental room for at least 3 d. All experimental procedures involving animals were approved by the Institutional Animal Care and Use Committee at the New York University Langone Medical Center and the Animal Advisory Committee at Zhejiang University.

**Behavioral assays**. All behavioral testing occurred between 0800 and 1800 h.

Spatial reference memory was tested using the MWM. A circular tank (diameter: 183 cm) was filled with water at 20 °C and surrounded by uniform blinds and quadrant-specific visual cues[55]. The water was made opaque by adding a sufficient amount of Tempera paint. Animal movement was tracked using a ceiling camera and Ethovision tracking software. Mice were introduced into the pool at pseudo-randomized drop locations outside of the target quadrant. During the "hidden platform learning task," a circular escape platform (15 cm diameter) was placed in the middle of a designated target quadrant 1 cm below the water surface. Mice were trained to find the platform by four 90 s trials per day, performed on a duty cycle of 14 min, for 4 consecutive days. During each block of trials, the mice were released from four pseudo-randomly assigned start locations (NW, SW, NE, and SW). A trial ended either when a subject rested on the hidden platform for 5 s or the end of the trial was reached. Mice that failed to find the platform by the end of the 90 s trial were manually guided to rest on the platform for 15 s. On day 5, a 90 s "probe trial" was conducted with the escape platform removed.

A modified radial arm maze was used to test working and reference memory at the same time[55]. The floor of the maze was made of white plastic, and the walls (13 cm high) consisted of transparent plexiglass. Each arm (13 × 40 cm) radiated from an octagonal central starting platform (diameter 23 cm) like the spokes of a wheel. The maze was elevated 75 cm above the floor and only four maze arms were baited. Each mouse was weighed daily throughout training and tested to monitor health and degree of food deprivation. Mice were deprived of food until their body weight was reduced to 85–90% of the initial level. Mice were acclimated to the food rewards (sweetened breakfast cereal) and food rewards were spread around the entire maze to encourage exploration (3 d). On subsequent days, food was placed only on the arms, then only at the ends of the arms before the test (2 d). After training, the food reward was placed at the end of only four arms of the radial arm maze before each test session. All doors were raised to allow the mouse to explore the maze completely and retrieve all food rewards (or until 10 min trial time was reached). The same four maze arms were baited each day and, across sessions, the mice generally learned to ignore the remaining four arms, which never contained a reward. Within a training session, re-entry into one of the four baited arms was considered a working memory error, while entry into a never-baited arm was considered a reference memory error.

For inhibitory avoidance training, a chamber was created consisting of a rectangular-shaped Perspex box divided into a safe compartment (white and illuminated) and a shock compartment (black and dark)[56,57]. Foot shocks were delivered to the grid floor of this chamber via a constant current scrambler circuit.

The two compartments were separated by an automatically operated sliding door. During training sessions, each mouse was placed in the safe compartment with its head facing away from the door. After 10 s, the door was automatically opened, allowing the mouse access to the shock compartment. The door closed after the mouse entered the shock compartment, and a brief foot shock (0.5 mA for 2 s with a 5 s delay) was administered to the mouse. Twenty-four hours after the training, the mouse was placed back in the safe compartment and its latency to enter the shock compartment was measured. The test was terminated at 300 s. 1 h after the test, the animal was heart-perfused and the brain was rapidly dissected for immunohistochemistry. Mice exposed to the inhibitory avoidance apparatus for the same duration as the trained animals without receiving a foot shock were used as the control group.

To assess locomotor activity, open field tests were performed in a novel environment under low-light conditions[55]. The open field chamber consisted of a square arena (40 × 40 cm) with opaque white walls. The test was initiated by placing mice in the middle of the open field and allowing them to move freely for 10 min while being tracked by the Ethovision automated tracking system. The chamber was cleaned with 70% alcohol between animals. Distance moved, velocity, and time spent in each predefined zone were analyzed.

To test for anxiety traits, we conducted an elevated zero maze test[55]. The zero maze consisted of an annular platform (diameter 40 cm, lane width 5 cm), elevated 40 cm from the floor, and was divided into four equal quadrants. Two opposite quadrants were enclosed by gray plastic walls (height 15 cm). In the test, mice were put on an open quadrant facing a closed quadrant and tracked for 8 min. Tests were videotaped and subsequently analyzed.

To assess muscle strength using all four limbs, the four limb-hanging test was performed in which a wire grid was used to measure the ability of mice to exhibit sustained limb tension to oppose their weight. The hang time is measured from the time the grid is inverted to the time that the mouse falls off the wire grid (determined visually and measured using a stopwatch) and the Holding impulse is the hang time multiplied by the body weight.

**Primary cultures of cortical and hippocampal neurons**. Hippocampal and cortical neurons were cultured from postnatal day 0 to 1 Sprague-Dawley rat pups or C57BL/6J mice. The hippocampus or frontal cortex were isolated and washed twice in ice-cold modified HBSS (4.2 mM $NaHCO_3$ and 1 mM HEPES, pH 7.35, 300 mOsm) containing 20% fetal bovine serum (FBS; Hyclone, Logan, UT) then three times with ice-cold HBSS only. Tissues were digested for 30 min in a papain solution (Worthington, NJ; 2.5 ml HBSS + 145 U papain + 40 μl DNase) at 37 °C with gentle shaking every 10 min. Digestion was stopped by adding 5 ml of modified HBSS containing 20% FBS. After additional washing, the tissue was dissociated using Pasteur pipettes of decreasing diameter. The cell suspension was pelleted twice and filtered with a 70 μm nylon strainer, and plated on 12 mm diameter coverslips coated with poly-D-lysine. The cultures were maintained in NbActiv4 (BrainBits, Springfield, IL). A 30% medium change was performed at 7 days, and once per week thereafter. Neurons were used 12–26 days after plating.

**Drug treatments and stimulation**. To induce CREB phosphorylation, $Ca^{2+}$/CaM/γCaMKII translocation, and other comparable or related events, we stimulated hippocampal and cortical neurons with the indicated high [$K^+$] solution or NMDA-containing solution at 37 °C for 60–600 s, and fixed the cells immediately after the stimulation (in 4% paraformaldehyde in PBS with 20 mM EGTA and 4% (w/v) sucrose). To induce c-Fos expression, we stimulated cells for 300 s, and then moved the coverslip back to the basal media for 40 min, at which point cells were fixed. 5 mM K Tyrode's consisted of (in mM): 150 NaCl, 5 KCl, 2 $MgCl_2$, 2 $CaCl_2$, 10 HEPES, 10 glucose, pH 7.4. When stimulating with elevated [$K^+$], $Na^+$ was adjusted to maintain osmolarity. All NMDA stimulation conditions were in 5 mM K Tyrode's containing 25 μM NMDA and 5 μM glycine. All $K^+$-rich and NMDA-containing stimulation solutions contained 0.5 μM TTX (Tocris) to block action potentials, unless otherwise indicated. Where indicated, drugs were added 30 min before and included throughout the stimulation: 10 μM NBQX (Tocris) to block AMPA receptors, 50 μM D-AP5 (Ascent Scientific) to block NMDA receptors, 10 μM Nimodipine (Abcam) to block $Ca_V$1 channels, 4 μM KN93 (Tocris) to block CaM Kinases, 50 μM PD98059 (Tocris) to block MEK1.

**Lentiviral transduction of cultured neurons**. To produce lentivirus, the pLVTHM shRNA or pCDH-EF1- γCaMKII constructs were transfected into 293FT cells along with the packaging plasmid psPAX2 and the envelope plasmid pMD2.g, kindly provided by Dr. D. Trono. After 16 h, the medium was changed. The supernatant was collected 24 h later and cleared of cell debris by filtering through a 0.45 mm filter. The viral particles were concentrated by centrifuging the filtrate at 70,000 × g for 2 h at 4 °C using a Beckman SW28 rotor. The viral pellet was then resuspended in sterile PBS, aliquoted, and stored at 80 °C. Lentivirus particles (0.5–1 ml of viral stock diluted in 20 ml of PBS per coverslip) were added to cultured neurons containing 500 μl of medium the day after plating. Twenty-four hours later, the cultures were fed with 1 ml of medium and used 4–5 days later, at which point infection efficiency was nearly 100%.

**Slice preparation**. Mice (postnatal 80–90 days) were anesthetized with isoflurane and the brain was quickly dissected out after decapitation. Transverse acute slices (350 μm thick) were sectioned on a vibratome in 95% $O_2$/5% $CO_2$ aerated ice-cold sucrose cutting solution (in mM: 206 Sucrose, 26 $NaHCO_3$, 11 Glucose, 2.5 KCl, 1 $NaH_2PO_4$, 20 $MgCl_2$, 0.5 $CaCl_2$). Then hippocampal slices were incubated in 95% $O_2$/5% $CO_2$ aerated ACSF (in mM: 122 NaCl, 3 KCl, 10 Glucose, 1.25 $NaH_2PO_4$, 1.3 $MgCl_2$, 2 $CaCl_2$, 26 $NaHCO_3$) at 32 °C to recover for 1 h, and then left at room temperature for 1 h for further recovery for use up to 5 h later.

**Electrophysiology**. Field excitatory post-synaptic potential (fEPSP) recordings were performed in area CA1 from 350 μm transverse acute slices. Recording was in an interface chamber at 32 °C, perfused with artificial cerebrospinal fluid (ACSF). Slices were allowed 20–30 min to equilibrate in ACSF before recording. Recordings were made with glass pipettes (3–5 MΩ) filled with ACSF and placed in the stratum radiatum of CA1. A concentric bipolar tungsten stimulating electrode was placed between area CA3 and CA1 for stimulation of the Schaffer collaterals. The intensity of the stimulating pulse was adjusted to produce responses 30–50% of the maximum and 30 min of baseline recording was performed to ensure the stability of responses. Late long-term potentiation (L-LTP) was induced by three 1 s trains of 100 Hz stimulation at 5 min intervals[5,24,32]. Stimulus intensity was equivalent in test and induction responses.

CA1 pyramidal cell membrane properties were determined with whole-cell current clamp recordings, performed with an intracellular solution containing (in mM): 130 K-Gluconate, 1 $MgCl_2$, 10 HEPES, 0.3 EGTA, 10 Tris-Phosphocreatine, 4 Mg-ATP, 0.3 Na-GTP and 0.1% biocytin. After maintenance of whole-cell recording for 15–20 min, slices were transferred to a 4% paraformaldehyde fixative solution overnight, then washed and stained with Alexa Fluor 488 streptavidin (1:700, Molecular Probes). Cells were imaged with a Zeiss confocal microscope and analyzed for dendritic morphology.

**Field stimulation and calcium imaging**. Following the cutting and recovery of hippocampal slices, they were incubated for 30 min at 37 °C, followed by 15 min at RT, in a 1:1 dilution of Fluo-4 (Life Technologies) in ACSF to load slices with the $Ca^{2+}$ indicator. Following loading, slices were placed on a stage, submerged in fresh oxygenated ACSF. Two platinum electrodes were positioned with a distance of ~10 mm on either side of the slices. The acute slices were field stimulated at 100 Hz for 5 s with square wave pulses (1 ms per pulse) at 15 V. Cells were imaged with a Zeiss LSM 510 meta Imager.M1 confocal microscope at 1.25 Hz. $\Delta F/F$ $Ca^{2+}$ responses were analyzed using Icy software.

**Perfusion and fixation of tissue**. Animals were anesthetized with a lethal dose of pentobarbital (100 mg/kg) and underwent intracardiac perfusion with 50 ml of ice-cold PBS. The brain was removed from the skull, the right side was immersed or perfused with ice-cold solution made up of 4% PFA in PBS, pH 7.4 and the left side was dissected into hippocampus and cortex that were snap-frozen and stored at −80 °C until they were used for biochemical analyses. Twenty-four hours PFA-fixed hemispheres were transferred to 30% sucrose until tissue sank, then OCT-embedded and stored at −80 °C until performance of further tissue sectioning and immunohistological analyses.

**Morphology analysis**. Six male mice (littermates aged P60-65, 3 mice per genotype) were used for morphological analysis. FD Rapid Golgi Stain Kit (FD Neu-roTechnologies, MD, USA) was used for the staining procedure, following all guidelines according to the manual. In brief, whole brains were dissected from each animal, rinsed in Milli-Q water, and immersed fully in prepared impregnation solution (solution A + B). The brains were stored at room temperature in the dark for 2 weeks before being transferred into solution C. Sections of 150 μm were cut on a vibratome, and mounted onto gelatin subbed slides (Southern Biotech, AL, USA). After the sections had dried (2–3 days, left at room temperature), the staining procedure described in the manual was followed. Hippocampal sections were imaged on a Zeiss AxioObserver Inverted light microscope with a ×60 lens. Between three and five neurons from hippocampal area CA1 were imaged from each animal, and fully traced utilizing Neurolucida software (Microbrightfield bioscience, VT, USA). The full extents of the apical and basal dendrites were traced and analysis for branch length and number and Sholl analysis were carried out with Neurolucida Explorer. Data was pooled for all neurons traced within individual genotypes, resulting in ten neurons being analyzed for each group.

**Viral injection surgery**. Mice were anesthetized with Isoflurane (0.5% oxygen/3% Isoflurane for induction and 0.5% oxygen/1.5% isoflurane for maintenance) during viral injections. After induction, mice were tested for deep anesthesia levels, defined as being unresponsive to noxious stimuli (i.e., ear twitch responses, pedal reflexes). A subcutaneous dose of buprenorphine (0.05–0.1 mg/kg) was injected and dextrose-lactated Ringer's solution (1.2 ml s.c.) was administered before the incision. After attainment of the appropriate level of anesthesia, the animal was secured using ear bars and ophthalmic ointment was applied. The scalp fur was shaved with a razor, the skin cleaned with betadine, and then a midline 1.5 mm incision was made with a sterile scalpel. The subcutaneous tissue was reflected away from the bone. The bone was dried with a cotton swab, and the stereotaxic

apparatus was used to mark the surface location of the coordinates for burr holes. 0.5-mm burr holes were drilled with a 23G1 needle over the parietal bone at the following locations bilaterally where later the injections were performed at four different points per hemisphere: 2.0–2.5 mm caudal and 1.4–1.7 mm lateral from the bregma. A glass pipette containing virus was attached to a Nanoject II system (Drummond Scientific company) and 100 nl of either lentivirus or saline was injected in each of the 4 holes per hemisphere. The injection microelectrode was slowly withdrawn 5 min after the virus infusion. Behavior experiments were performed at least 14 days after surgery. The effectiveness of virus infections was confirmed by staining with antibody against HA (Supplementary Fig. 6f).

After the virus injections, the incision was closed with silk sutures and topical antibiotic was applied to the top of the suture to prevent infection. During anesthesia recovery, mice were observed every 10–60 min and body temperature was maintained at 37 °C using a heated pad until they were placed in their home cage. All mice were maintained under a 12:12 h light:dark cycle after that and food and water were provided ad libitum. Mice thrived and had normal activity within 12 h after the injections. Post-surgical anesthesia: a subcutaneous injection of buprenorphine (0.05–0.1 mg/kg) at 24 h post-surgery was administered as well as a dose of Dextrose-lactated Ringer's solution (1.2 ml s.c.). Animals were monitored twice daily for 3 more days for signs of pain or distress. No signs of infection and no mortality were present for any of the animals in the study.

**Protein expression and purification**. The wild-type and mutant γCaMKII and CaM were expressed in HEK 293 Cells and E. coli, respectively (CaM and CaM (C75) were kind gifts from Dr. Neal Waxham). The CaMKII proteins were purified using Calmodulin Sepharose affinity beads (Calmodulin Sepharose 4B, GE Healthcare) and CHT ceramic hydroxyapatite (Bio-rad, #1588000) with gravity flow. For CaM and its mutated counterpart CaM(C75), the proteins were purified with UNO-sphere Q anion exchange resin (Bio-rad, #1560101) and CHT ceramic hydro-xyapatite (Bio-rad, #1588000) with gravity flow. All proteins were purified following the standard protocol from the manufacturer and eluted with a step gradient of increasing concentrations of potassium phosphate buffer, as confirmed with gel electrophoresis before the appropriate fractions were pooled.

**Measurement of CaM dissociation rate from CaMKII**. CaM(C75) was labeled with IAEDANS (Thermofisher) as previously described with small modifications[58]. Briefly, 50 nM CaM(C75) was incubated with 0.5 mM IAEDANS (urea not present) overnight followed by dialysis using PD MiniTrap™ G-25 against 50 mM MOPS, pH 7.0, 150 mM KCl, 0.1 mM EGTA, 0.5 mM $CaCl_2$ and 0.1 mg/ml BSA. 1 mM MgATP was added into the reaction solution containing 0.5 mM $Ca^{2+}$, CaM (C75)$_{IAE}$ and γCaMKII or γCaMKII R292P to enable phosphorylation at Thr 287. After incubation at room temperature for 5 min, a 75-fold excess of unlabeled CaM was added to exchange with labeled CaM(C75)$_{IAE}$. For fluorescence measurements, excitation was at 345 nm and emission at 465 nm was monitored at 3 Hz using a Flex station. The experimental data was fitted with a two phase decay function

$$Y = A1*\exp(-t/\tau_1) + A2*\exp(-t/\tau_2) + Y_0).$$

**CaMKII activity assay**. The kinase activity of γCaMKII was measured using the CyclLex Calmodulin kinase II Assay Kit (Cat # CY-E1173, CycLex). Briefly, purified γCaMKII or γCaMKII R292P (equal amounts) was added to the kinase reaction buffer provided with the kit and incubated for 30 min at 30 °C in 96-well plates. Wells were washed for five times with wash buffer and 100 μl of the horseradish peroxidase-conjugated detection antibody MS-6E6 was added. After 60 min incubation at room temperature, samples were washed five times, and the substrate reagent (100 μl) was added. After 15 min incubation at room temperature, the reaction was stopped by the addition of a stop solution and the absorbance read at 450 nm with a microplate reader.

**Immunocytochemistry**. Cultured cells were fixed in ice-cold 4% paraformaldehyde (PFA) in PBS supplemented with 20 mM EGTA and 4% (w/v) sucrose. Fixed cells were then permeabilized with 0.1% Triton X-100, blocked with 6% (cells) or 10% (10 μm hippocampal slices) normal donkey serum, and incubated overnight (cells) or two overnights (10 μm hippocampal slices) at 4 °C in primary antibodies: rabbit anti-pCaMKIV (1:500, sc-28443-R, Santa Cruz Biotechnology); mouse anti-CaM (1:500, 05-173, Millipore); rabbit anti-pCREB (1:333, 9198, Cell Signaling Technology); rabbit anti-c-Fos (1:500, 2250, Cell Signaling Technology); rabbit anti-BDNF (1:10,000, sc-546, Santa Cruz); rabbit anti-Arc (1:1000, sc-17839, Santa Cruz); mouse anti-HA (1:10,000, MMS-101P, Covance); mouse anti-HA (1:1000, H9658, Sigma-Aldrich); goat anti-Rsk2 (1:100, sc-1430, Santa Cruz); rabbit anti-CRTC1 (1:100, A300-769A, Bethyl), mouse anti-αCaMKII (1:2000, 13-7300, ThermoFisher). The next day, cells were washed with PBS, incubated at room temperature for 45 min (for cells) or 2 h (hippocampal slices) with Alexa-tagged secondary antibodies (1:1000 for cells, 1:500 for hippocampal slices, Molecular Probes), washed with PBS and mounted using ProLong Gold (Invitrogen). The cells or slices were imaged with a ×40 or ×60 (1.3 NA) oil objective on a confocal microscope LSM 510 (Carl Zeiss, Inc.) or an Axioplan epifluorescent microscope equipped with an AxioCam digital camera using AxioVision.

**Western blot**. SDS-PAGE loading buffer and β-mercaptoethanol were added to the lysate (25 and 10% of the total lysate volume, respectively), and the mixture was heated to 95 °C for 5 min. Cellular protein was separated on a 10% SDS-PAGE gel and transferred to Immobilon transfer membrane (Millipore). The membrane was then blocked at room temperature for 1 h in Odyssey blocking buffer (Li-Cor Biosciences, Lincoln, NE). The membrane was incubated overnight with antibodies as the γCaMKII antibody (rabbit, 1:1000) was raised against γCaMKII (amino acids 441–460) and its specificity was tested in γCaMKII$^{-/-}$ mice; rabbit anti-Lamin B (1:1000, 12586, Cell Signaling Technology); rabbit anti-GAPDH (1:1000, 5174, Cell Signaling Technology); mouse anti-HA (1:1000, H9658, Sigma); mouse anti-CaM (1:500, 05-173, Millipore); rabbit anti-c-Fos (1:500, 2250, Cell Signaling Technology); rabbit anti-BDNF (1:10,000, sc-546, Santa Cruz); rabbit anti-Arc (1:1000, sc-17839, Santa Cruz); mouse anti-αCaMKII (1:2000, 13-7300, Thermo-Fisher); mouse anti-βCaMKII (1:1000, 139800, invitrogen); rabbit anti-αCaMKII (1:1000, 13-7300, abcam); rabbit anti-pPKA (1:500, sc-377575, Santa Cruz, to track PKA activation); rabbit anti-p44/42 MAPK (1:1000, 4370, Cell Signaling Technology); rabbit anti-p38 MAPK (1:1000, 9211, Cell Signaling Technology); rabbit anti-NKCC1 (1:200, AB3560P, Chemicon International); rabbit anti-KCC2 (1:200, 07-432, Chemicon International); rabbit anti-GABA$_A$R (1:200, Chemicon International); rabbit anti-GluR (1:500, Millipore; Rabbit). The specificity of BDNF antibody for WB or ICC was tested previously[59–61] and the band (~16 kDa) was examined for WB. The following day, the membrane was washed with 0.1% Tween 20 in PBS and incubated with an IRDye-labeled secondary antibody (Li-Cor) for 1 h. After incubation, the membrane was washed with PBS and imaged with Li-Cor Odyssey imaging system. All the bands were analyzed with Image Studio. For nuclear extraction, the nuclear proteins of cortical neurons (DIV13-15) were isolated with NE-PER Nuclear and Cytoplasmic Extraction Reagent Kit (Thermo Scientific) before the gel electrophoresis[19]. For CaM overlay, CaM (Enzo Life Sciences) was added at a final concentration of 0.5 μg/ml in the 5 mM K$^+$ Tyrode's and incubated with membrane for 1 h before addition of the primary antibody for CaM[62]. Images in the main text have been cropped for presentation. Full size images are presented in Supplementary Figs. 7–9.

**Image analysis**. For cultured neurons, images for protein quantification were analyzed using a custom-written macro in ImageJ (NIH). The nuclear marker DAPI and an antibody against the neuron-specific protein MAP2 were used to delineate the nucleus and cytoplasm, respectively. A region of interest adjacent to each neuron analyzed was used as an 'off-cell' background. In cortical cultures, αCaMKII was used as a marker to restrict analysis to excitatory neurons. The protein level in each neuron was calculated by subtracting the average intensity in the 'off-cell' background region of interest from the average intensity in the appropriate region of interest for each neuron. The nuclear:cytoplasmic ratio was calculated by comparing nuclear immunoreactivity with apical dendrite immunoreactivity (immunoreactivity of the membrane profile was not included).

For hippocampal slices, two images were taken in the CA1 from every slice (two slices per animal). Using ImageJ, ROIs were chosen at each DAPI + nucleus in a given section. The average pixel intensity of staining for the signal-of-interest was obtained within the boundary of each nuclear ROI. From this value, a background value (taken in the stratum radiatum) was subtracted. For c-Fos staining, an arbitrary threshold of 50 intensity units was set and applied to all slices, and the percentage of cells with c-Fos staining >50 was calculated for each image.

**DNA constructs**. The point mutation was produced using QuikChange Lightning Site Directed Mutagenesis Kit (Agilent) with pCDH-EF1a-HA-γCaMKII[19] as template and the following primer and its reverse-complement (mutated bases in lower case bold):

γCaMKII R292P: G AAG TCG GAT GGC GGT GTC AAG AAA AGG AAG **ga**G AGT TCC AGC GTG CAC CTA ATG GAG CC

All γCaMKII constructs in this study utilized the human γ$_A$·CaMKII isoform of γCaMKII.

**Data availability**. The data that support the findings of this study are available from the corresponding author upon reasonable request.

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

## Acknowledgements

We thank Drs. Eric Olson and Johannes Backs for generously providing γCaMKII exc-KO and γCaMKII KO mice. We thank Dr. Neal Waxham for providing CaM and CaM (C75) constructs. We thank Howard Schulman for strategic insights, and Gyuri Buzsaki, Orrin Devinsky, Gord Fishell, Adam Mar, and Nicholas Stavropoulos for comments on earlier versions of the manuscript. We also thank Guoling Tian and Hongyan Shi for technical assistance, Boxing Li and other Tsien lab members for advice. This work was supported by research grants to H.M. (the National Natural Science Foundation of China (31771109 and 31722023), the Fundamental Research Funds for the Central Universities of China (2018XZZX002-02), 111 project (B13026), and the K.C. Wong Education Foundation) and R.W.T. (NIH support from NIGMS (GM058234), NINDS (NS24067), the NIMH (MH071739), and the Druckenmiller, Simons, Mathers, and Burnett Family Foundations). S.M.C. is supported by a Medical Scientist Research Service Award (T32GM007308). A.S. is supported in part by FACES.

## Author contributions

H.M. and R.W.T. conceived the project. S.M.C. performed calcium imaging, immunostaining for rescuing experiments in WT and exc-KO mice, and cell culture experiments related to NMDA signaling. B.S. performed LTP recordings and N.N.T. performed other electrophysiology experiments. X.H. did IA experiments and immunostaining after the tests. N.N.T. and C.M. performed the experiments to assess neuronal morphology. A.S. performed viral injections. H.M. performed all the other experiments with the assistance of Y.W., G.Z., S.W., X.H., I.K., and S.S., H.M., S.M.C., and R.W.T. wrote the manuscript.
