## [Peer Review File · Nature Communications]

Editorial Note: Several images have been redacted as indicated to protect copyright claims. These images can be found in the publications in which they originally appear, indicated at the place of redaction.

Reviewers' comments:

Reviewer #1 (Remarks to the Author):

In this manuscript the authors extend significantly upon their previously articulated model of nuclear calmodulin shuttling as a mechanism for synapse to nucleus transcriptional signaling by showing the importance of this process for physiological relevant models of synaptic plasticity and learning and memory. Previously the Tsien lab identified nuclear shuttling of calmodulin, and gamma CaMK2 as the shuttle mechanism, as key factors mediating the calcium dependent activation of the transcription factor CREB in neurons. Here they first use Camk2g knockout mice and show they have impaired learning and memory in a series of behavioral tasks, impaired induction of CREB target genes, and impaired LTP in hippocampal slice. They then do a series of pharmacology experiments to support their argument that these changes are due to impaired calmodulin transfer to the nucleus and CaMKIV activation of CREB. These data are solid, however the most exciting findings of the manuscript involve a mutation of gamma CaMKII that was reported in a child with intellectual disability. The authors show data that characterizes this mutation as selectively impairing the ability of gamma CaMKII to trap Ca/CaM and transport it to the nucleus. They then perform a series of rescue experiments in the gamma CaMKII null background and show that only WT but not R292P mutant gamma CaMKII can rescue nuclear CaM, Fos induction, LTP and learning behavior. These data provide novel and compelling support for the CaM trapping model of gamma CaMKII function and as thus substantially expand understanding of the mechanisms of activity-inducible transcription that are important for synaptic plasticity and behavioral adaptation.

The experiments overall are performed with technical excellence, and I have only one minor concern, which is the use of the BDNF antibody. Unlike Fos and Arc, for which commercial antibodies have been well validated by knockout approaches for their specificity, the commercial BDNF antibodies have rarely been validated. BDNF is highly conserved across species making it hard to generate high antibodies, and it is well known among researchers in the field that the signals detected by many of these commercial antibodies on western or brain section staining do not go away when tested on knockout. For this reason, if the authors wish to include BDNF immunostaining or western blotting in their data, they should validate the commercial antibody they are using with knockout or knockdown neuronal samples.

Reviewer #2 (Remarks to the Author):

The authors probe the function of γ CaMKII in learning and memory, postulating that the kinase acts as a shuttle for the nuclear translocation of Ca/CaM, which is in turn required for

long term plasticity and learning. The study includes in-vivo behavioral results paired with in-vitro testing of the hypothesized mechanism, making for a compelling and high quality study.

Major comments

1. The authors put forth a hypothesis whereby a major role for γ CaMKII in spatial learning is as a shuttle for Ca/CaM entry into the nucleus. However, the robust dominant negative effect of the R292P γ CaMKII does not appear to directly fit with this hypothesis. While there are multiple possible explanations for this, they have not been explored or discussed. In particular, it would be useful to know the approximate ratio of mutant vs. WT γ CaMKII in the experiment. Additionally, does introduction of exogenous R292P γ CaMKII into WT neurons inhibit Ca/CaM translocation in culture?

2. The pharmacology in Fig. 2 h-k would benefit from application of the drug in the absence of NMDA stimulation. It is difficult to determine if the drugs are specifically blocking the effect of NMDA.

Minor comments

1. The data in Fig. 2b is displayed as exemplar images, with quantification presented in c1 for trained animals, while untrained animals are quantified somewhat differently in extended fig. 4 (the scale for nuclear CaM is different between the main text and online material – should the x-axis for main text Fig 2c1 be CaM nuc/cyt ratio?). This makes it difficult to do a simple comparison of all the data. It would be helpful if the data could be plotted together, either in the main text or the supplement.

2. Fig. 2c1 does not appear to be referenced in the main text.

3. Fig. 3b, the peak value for the KO and the KO+R292P appear to be different – does this reach a level of significance?

4. Fig. 3b, it appears as if the stimulus arrows may be slightly shifted on the x-axis.

5. The paper would benefit from an expanded introduction and discussion.

Reviewer #3 (Remarks to the Author):

The authors group previously reported that CaMKII-gamma shuttles Ca²⁺/CaM between cytosol and nucleus. Ca²⁺/CaM, in turn, activates CaMKIV-CREB pathway. This leads to the induction of number of activity-induced gene products such as c-fos and Arc necessary for long-term synaptic plasticity. The current study is a follow-up of the previous study by using a CaMKII gamma knockout animal as well as newly identified mutation found in human patient with intellectual disability.

The critical issue is whether one can separate the catalytic role of CaMKII-gamma versus the shuttling role by using the mutation. This reviewer feels that this point is weak in this

paper. More specifically, the kinase activity of R292P mutant must be more thoroughly tested to rule out the effect of mutation on kinase activity. The data in Figure 3 c-e, and g are not much meaningful, certainly cannot estimate the binding affinity to CaM or specific activity of the enzyme. V_{max} , affinity to CaM, ATP and substrate must be tested. Also, the autophosphorylation (time course etc) and resultant constitutive activity must be also tested. Given that the CaM dissociates quickly, it is possible that overall kinase activation process is impaired. I agree that the authors used NLS-Nrgn-HA to test the sufficiency of releasing CaM to induce c-fos under 40 mM K⁺. But the authors has not tested it in intact animal. Phosphoproteome of non-nuclear fraction (for example PSD) might be an approach to show if there is any alteration in kinase function or not.

The rest of studies are overall carefully conducted and if the point above is cleared, this paper will be a good candidate for publication.

Minor comments.

I see some problem in data presentation and analysis.

1. Figure 1B. Show original blot.
2. Figure 2B. KO c-fos staining. I cannot see how the authors defined positive cells given poor S/N ratio.
3. Figure 2D-2K. Show representative staining.
4. Figure 3B. WT group. The baseline is going up. Go back to the original data and reject data showing unstable baseline. Also, why E-LTP phase is different between WT rescue group vs KO?
5. Figure 3h-j. Show representative raw data. WT and R292P rescue need no stim level for each measurement.
6. Figure 3k. NLS-Nrgn (WT or S36D)-HA only group is needed. Also, no stimulation groups is needed, at least for NLS-Nrgn (WT)-HA.
7. Figure 4b. Same as comment 2.
8. Figure 4d. X-axis. What is the unit?
9. Extended Figure 1e. Show original blot.

Reviewer #1 (Remarks to the Author):

In this manuscript the authors extend significantly upon their previously articulated model of nuclear calmodulin shuttling as a mechanism for synapse to nucleus transcriptional signaling by showing the importance of this process for physiological relevant models of synaptic plasticity and learning and memory. Previously the Tsien lab identified nuclear shuttling of calmodulin, and gamma CaMK2 as the shuttle mechanism, as key factors mediating the calcium dependent activation of the transcription factor CREB in neurons. Here they first use Camk2g knockout mice and show they have impaired learning and memory in a series of behavioral tasks, impaired induction of CREB target genes, and impaired LTP in hippocampal slice. They then do a series of pharmacology experiments to support their argument that these changes are due to impaired calmodulin transfer to the nucleus and CaMKIV activation of CREB. These data are solid, however the most exciting findings of the manuscript involve a mutation of gamma CaMKII that was reported in a child with intellectual disability. The authors show data that characterizes this mutation as selectively impairing the ability of gamma CaMKII to trap Ca/CaM and transport it to the nucleus. They then perform a series of rescue experiments in the gamma CaMKII null background and show that only WT but not R292P mutant gamma CaMKII can rescue nuclear CaM, Fos induction, LTP and learning behavior. These data provide novel and compelling support for the CaM trapping model of gamma CaMKII function and as thus substantially expand understanding of the mechanisms of activity-inducible transcription that are important for synaptic plasticity and behavioral adaptation. The experiments overall are performed with technical excellence, and I have only one minor concern, which is the use of the BDNF antibody. Unlike Fos and Arc, for which commercial antibodies have been well validated by knockout approaches for their specificity, the commercial BDNF antibodies have rarely been validated. BDNF is highly conserved across species making it hard to generate high antibodies, and it is well known among researchers in the field that the signals detected by many of these commercial antibodies on western or brain section staining do not go away when tested on knockout. For this reason, if the authors wish to include BDNF immunostaining or western blotting in their data, they should validate the commercial antibody they are using with knockout or knockdown neuronal samples.

We thank the reviewer for the very thoughtful and positive comments. The reviewer is definitely right that there are some issues in the field about the specificity of various anti-BDNF antibodies. In our case, the specificity of this antibody (SC-546 or N20, Santa Cruz) has been discussed in a great detail (Rantamaki et al., 2013), and has been validated via both immunostaining (Braun et al., 2017) and western blot (Matsumoto et al., 2008) using brain samples of BDNF KO mice (see the figure below). Interestingly, just as the reviewer points out, even in the images below (Matsumoto et al., 2008), there were some bands (~38 kDa) that didn't go away in the BDNF KO samples. Importantly, however, the band marked by the red rectangle did disappear in the samples of BDNF KO mice, and the band we have examined in this study is at the

same molecular weight (supplemental Fig. 1g; for convenience, repeated here below other panels, with the molecular weight standard shown). Given the agreement in regard to molecular weight, the change in BDNF levels can be reliably examined using Western blot analysis with this antibody. Nonetheless, we have modified the methods section of the revised manuscript to further acknowledge and address this concern.

[Redacted: Fig. 2 of Braun *et al.*, 2017, PMID: 28266222]

Matsumoto *et al.*, 2008

Supp. Fig. 1g

Reviewer #2 (Remarks to the Author):

The authors probe the function of γ CaMKII in learning and memory, postulating that the kinase acts as a shuttle for the nuclear translocation of Ca/CaM, which is in turn required for long term plasticity and learning. The study includes in-vivo behavioral results paired with in-vitro testing of the hypothesized mechanism, making for a compelling and high quality study.

We thank the reviewer for the positive comments about our combination of *in vivo* and *in vitro* approaches. It is gratifying that the reviewer views our work as a compelling and high-quality study.

Major comments

1. The authors put forth a hypothesis whereby a major role for γ CaMKII in spatial learning is as a shuttle for Ca/CaM entry into the nucleus. However, the robust dominant negative effect of the R292P γ CaMKII does not appear to directly fit with this hypothesis. While there are multiple possible explanations for this, they have not been explored or discussed. In particular, it would be useful to know the approximate ratio of mutant vs. WT γ CaMKII in the experiment. Additionally, does introduction of exogenous R292P γ CaMKII into WT neurons inhibit Ca/CaM translocation in culture?

The reviewer brings up excellent points! In fact, we were quite intrigued to see a robust dominant negative effect of γ CaMKII R292P, the mutation identified in an ID patient, in multiple experimental systems. Following the reviewer's suggestions, we have analyzed brain samples with virally overexpressed variants of human γ CaMKII in KO mice and compared them with WT mice. Our results indicate that levels of γ CaMKII were 3.5-fold or 3.8-fold higher than WT in KO mice expressing γ CaMKII WT or γ CaMKII R292P, respectively (Extended Data Fig. 6e). Moreover, following through on the reviewer's suggestion, we found that overexpressing exogenous γ CaMKII R292P in WT cultured neurons did inhibit Ca^{2+} /CaM translocation (now illustrated in Extended Data Fig. 6g). This key experiment now provides further support for the idea that γ CaMKII R292P can work in a dominant negative fashion to regulate nuclear signaling. As the reviewer correctly points out, there may be several explanations for this effect, building on our previous elucidation of the multiple checkpoints in γ CaMKII-mediated transport of CaM. The dominant negative phenotype could arise from competition for: (1) cross-phosphorylation by α/β CaMKII (if fewer WT molecules undergo cross-phosphorylation at T287, less CaM would be trapped and delivered to the nucleus), (2) CaN-mediated dephosphorylation of γ CaMKII at S332 to unmask the nuclear localization sequence, or (3) binding to players involved in nuclear import such as importin molecules. We now provide a brief consideration of these various interpretations of the dominant negative effect in the Discussion section.

2. The pharmacology in Fig. 2 h-k would benefit from application of the drug in the absence of NMDA stimulation. It is difficult to determine if the drugs are

specifically blocking the effect of NMDA.

We thank the reviewer for suggesting these control experiments. We have added data on CaM localization and pCREB, pCaMKIV, and c-Fos nuclear values in the absence of NMDA stimulation, now incorporated in Extended Data Fig. 5a. Our results suggest that the drugs themselves have no effect on nuclear events, and that they do indeed specifically block the effect of NMDA.

Minor comments

1. The data in Fig. 2b is displayed as exemplar images, with quantification presented in c1 for trained animals, while untrained animals are quantified somewhat differently in extended fig. 4 (the scale for nuclear CaM is different between the main text and online material – should the x-axis for main text Fig 2c1 be CaM nuc/cyt ratio?). This makes it difficult to do a simple comparison of all the data. It would be helpful if the data could be plotted together, either in the main text or the supplement.

In both Fig. 2c1 and extended Fig. 4, the X-axis refers to nuclear CaM level. In each of these experiments, nuclear CaM values in trained WT and exc-KO animals (Fig. 2c1), or trained WT and exc-KO animals injected with saline or virus (Fig. 4) were normalized to CaM values in naïve untrained controls (grey dotted line). The reviewer noticed an error in the x-axis label for Fig. 4c that has since been corrected. We agree that it would have been helpful if we were able to plot the data together. However, the two sets of experiments (Fig. 2 and Fig. 4) were performed by different people in the lab, with different variations of the immunostaining protocol. It is satisfying that even though the experiments were performed using different protocols in different hands, nuclear CaM was always positively correlated to c-Fos in an activity-dependent manner. However, since differences such as antibody incubation time also change the signal detection sensitivity between two sets of experiments, the experimental conditions preclude the possibility of a direct comparison using a combined plot. For this reason, we did not combine the plots, but we have checked the text to make sure that we did not draw any conclusions by comparing these two sets of experiments.

Fig. 2c1 does not appear to be referenced in the main text.

We thank for reviewer for spotting this oversight, which has since been corrected.

3. Fig. 3b, the peak value for the KO and the KO+R292P appear to be different – does this reach a level of significance?

Yes, there is a 26% increase with the expression of R292P for the peak value, with a p value of 0.04. It may be caused by the effect of γ CaMKII as a kinase during E-LTP if virally overexpressed. Indeed, we regard this as *in situ*, albeit circumstantial, evidence for the efficacy of overexpressed γ CaMKII as a phosphorylating enzyme, working alongside the endogenous kinase (α CaMKII).

4. Fig. 3b, it appears as if the stimulus arrows may be slightly shifted on the x-axis.

We thank the reviewer for this observation. The stimulus arrows in Fig. 3b have been corrected in the revised version.

5. The paper would benefit from an expanded introduction and discussion.

We have expanded the introduction and discussion in the revised version. Thanks for opening up this opportunity to make our story clearer to the general reader.

Reviewer #3 (Remarks to the Author):

The authors group previously reported that CaMKII-gamma shuttles Ca²⁺/CaM between cytosol and nucleus. Ca²⁺/CaM, in turn, activates CaMKIV-CREB pathway. This leads to the induction of number of activity-induced gene products such as c-fos and Arc necessary for long-term synaptic plasticity. The current study is a follow-up of the previous study by using a CaMKII gamma knockout animal as well as newly identified mutation found in human patient with intellectual disability. The critical issue is whether one can separate the catalytic role of CaMKII-gamma versus the shuttling role by using the mutation. This reviewer feels that this point is weak in this paper. More specifically, the kinase activity of R292P mutant must be more thoroughly tested to rule out the effect of mutation on kinase activity. The data in Figure 3 c-e, and g are not much meaningful, certainly cannot estimate the binding affinity to CaM or specific activity of the enzyme. V_{max}, affinity to CaM, ATP and substrate must be tested. Also, the autophosphorylation (time course etc) and resultant constitutive activity must be also tested. Given that the CaM dissociates quickly, it is possible that overall kinase activation process is impaired. I agree that the authors used NLS-Nrgn-HA to test the sufficiency of releasing CaM to induce c-fos under 40 mM K⁺. But the authors has not tested it in intact animal. Phosphoproteome of non-nuclear fraction (for example PSD) might be an approach to show if there is any alteration in kinase function or not.

We thank the reviewer for the constructive criticism. We were not attempting to “separate the catalytic role of CaMKII-gamma versus the shuttling role by using the mutation.” This point was much more directly addressed in our previous work¹, recapitulated below for the convenience of the editor (labeling of panels D-F as in 2014 *Cell* paper). The classical K43R modification renders the γ CaMKII kinase-dead, yet its expression gives identical results to the catalytically active protein. Kinase-dead and catalytically active CaMKII match up well in undergoing translocation to the nucleus (D), supporting CaM redistribution (E), and mediating CREB phosphorylation (F). In separate experiments in the 2014 paper and in this MS, we used direct delivery of nuclear Ca²⁺/CaM via NLS-Nrgn to trigger CREB phosphorylation even after γ CaMKII had been knocked down or replaced with trapping-defective γ CaMKII. This set of findings was taken as evidence for the importance of nuclear Ca²⁺/CaM delivery. We agree with the reviewer that our current experiments with R292P γ CaMKII, while consistent with the published data, do not by themselves provide evidence for excluding a catalytic role; once again, our reasoning relied on the direct kinase-dead experiment. The Discussion has been rewritten to make this point more clear.

[Redacted: Fig. 6 of Ma et al., 2014, PMID: 25303525]

We presume that the reviewer's interest in knowing how the R292P mutation affects other functions of γ CaMKII, such as V_{\max} , affinity for CaM, ATP and substrates, and the rate of autophosphorylation and activation all stems from his/her supposition that changes in catalytic activity could be involved. Consideration of the literature sets bounds on what might reasonably be expected for effects of the R292P mutation in γ CaMKII. We discussed this with Howard Schulman, and he pointed us to the K291E mutation in α CaMKII, reported in Yang and Schulman (*JBC* 1999). They showed that K291 pairs up with E105, almost certainly by electrostatic interaction: while charge reversal at either site enhances affinity for $\text{Ca}^{2+}/\text{CaM}$ by 5-fold, presumably by destabilization of the basal CaM-free state, affinity for $\text{Ca}^{2+}/\text{CaM}$ in a double mutant is almost perfectly reverted to normal. For the convenience of the reviewer these data are summarized in the table below.

[Redacted: Table 1 of Yang et al., 1999, PMID: 10473573]

Both by primary sequence and 3-D structure, α - and γ CaMKII are highly homologous in the region surrounding the mutation. Thus, in neutralizing the charge at position 292, the R292P mutation in γ CaMKII would also be expected to destabilize the basal CaM-free state and thus enhance $\text{Ca}^{2+}/\text{CaM}$ binding rather than oppose it.

In our opinion, the kinase-dead result for CREB phosphorylation renders moot considerations of how catalytically active γ CaMKII might be under physiological conditions of $\text{Ca}^{2+}/\text{CaM}$ elevation. Nevertheless, we are interested to test whether changes in its properties as a kinase have influence on functions other than signaling to nuclear CREB. Indeed our lab is making a γ CaMKII kinase-dead transgenic mouse to look for possible catalytic roles of γ CaMKII over and above its function as a shuttle. We contend that this line of research is best left for a future study.

We agree that investigating the phosphoproteome of a non-nuclear fraction would have been a telling result if the kinase activity of γ CaMKII had been demonstrably

important in mediating E-T coupling. As our previous data straightforwardly demonstrated that a kinase-dead γ CaMKII K43R mutation had no bearing on E-T coupling¹, these experiments were not performed. However, the reviewer's comments alerted us to consider possible effects of γ CaMKII as a kinase, described in detailed point 4 below.

The rest of studies are overall carefully conducted and if the point above is cleared, this paper will be a good candidate for publication.

We thank the reviewer for all of their critical suggestions. In light of the ability of the kinase-dead γ CaMKII mutant to fully support E-T coupling, we have conducted additional experiments to test the dominant negative effects of R292P. Furthermore, in our revised manuscript we expanded our discussion of other possible eventualities of the R292P mutation.

Minor comments.

I see some problem in data presentation and analysis.

1. Figure 1B. Show original blot.

The original blots have been added to Supplementary Figures 1e and 1g in the revised manuscript.

2. Figure 2B. KO c-fos staining. I cannot see how the authors defined positive cells given poo[r] S/N ratio.

In order to define c-Fos positive cells, we set an arbitrary threshold of 50 arbitrary units, and we held this standard for all conditions. As is demonstrated in the figure below, the majority of cells have a c-Fos value below 50, with a small number of cells exhibiting much higher levels. The mean nuclear value of c-Fos is also significantly higher in WT vs KO (not shown). However, given the nature of c-Fos as an immediate-early gene whose expression is activated, we felt that selecting nuclear values that surpassed a certain threshold better captured the value of nuclear c-Fos levels.

3. Figure 2D-2K. Show representative staining.

We thank the reviewer for the suggestion and have included representative staining for Fig. 2d-k. The images can be found in Extended Data Fig. 5b,c.

4. Figure 3B. WT group. The baseline is going up. Go back to the original data and reject data showing unstable baseline. Also, why E-LTP phase is different between WT rescue group vs KO?

Based on the reviewer's suggestion, we have omitted one experiment from the WT group that had an obviously unstable baseline. The revised pooled data now displays a respectably stable baseline and shows differences between the WT and R292P rescue groups—thanks for the suggestion.

The reviewer is correct to point out that the E-LTP phase is different between KO+WT rescue group vs KO+sham. One possible interpretation is that the exogenous γ CaMKII is overexpressed sufficiently to reveal its efficacy as a protein kinase, working alongside the endogenous phosphorylating enzyme (α CaMKII). Interestingly, the R292P γ CaMKII variant also enhances E-LTP as reflected by field EPSP slope immediately after induction ($p > 0.14$ for R292P γ CaMKII vs γ CaMKII) but the effect of R292P is much more transient, as expected if its catalytic activity were to decline more rapidly than WT due to lack of CaM trapping. The overall pattern is consistent with the idea that catalytic activity of R292P γ CaMKII is intact; the evidence, while admittedly circumstantial, comes from *in situ* physiology and might provide a useful clue to enzymatic efficacy.

5. Figure 3h-j. Show representative raw data. WT and R292P rescue need no stim level for each measurement.

The representative raw data was actually shown in Fig 3g, and directly correspond to the histogram bars in Fig 3h-j. Now, in response to the comment, we also show results in the absence of stimulation, comparing WT and the KO with R292P rescue; the representative raw data and pooled results are now presented in Extended Data Fig. 6b. For Figure 3 we chose to keep the Western blot lanes restricted to those needed to demonstrate the key points of this experiment for the sake of clarity.

6. Figure 3k. NLS-Nrgn (WT or S36D)-HA only group is needed. Also, no stimulation groups is needed, at least for NLS-Nrgn (WT)-HA.

We have performed related control experiments in our previous work¹, reproduced here for the convenience of the reviewers and editor. In the absence of stimulation, overexpressing NLS-Nrgn-HA didn't affect nuclear signaling (marked by the green rectangle). With stimulation, overexpressing NLS-Nrgn or S36D itself also didn't affect nuclear signaling (marked by the red rectangle).

[Redacted: Fig. S7 of Ma *et al.*, 2014, PMID: 25303525]

7. Figure 4b. Same as comment 2.

As described in the response to comment 2, an arbitrary threshold was set to define cells as c-Fos+ or c-Fos-.

8. Figure 4d. X-axis. What is the unit?

The x-axis in Fig. 4d and Fig. 4e is nuclear CaM value, normalized in both figures to exc-KO+saline. The x-axis has been revised in the current version.

9. Extended Figure 1e. Show original blot.

The original blot is now shown in the revised version of Extended Data Fig. 1e.

We thank the reviewer for the careful attention to detail. We believe that the experimental responses have made the paper stronger and hope that our replies to the penetrating questions are clear enough.

1. Ma H, *et al.* gammaCaMKII shuttles Ca(2)(+)/CaM to the nucleus to trigger CREB phosphorylation and gene expression. *Cell* **159**, 281-294 (2014).

REVIEWERS' COMMENTS:

Reviewer #1 (Remarks to the Author):

The authors have addressed all of my concerns, and I think that they have done an exemplary job of addressing the concerns raised by the other reviewers. The combination of data shown in this manuscript are compelling and interesting, deepening our understanding of excitation-transcription coupling.

Reviewer #2 (Remarks to the Author):

The authors have fully addressed all comments and have provided a high quality paper which will be of significant interest to the field.

Reviewer #3 (Remarks to the Author):

The authors adequately revised the manuscript. I totally understand that the first point (kinase vs carrier role) has been already addressed in their earlier paper.

Reviewer #1 (Remarks to the Author):

The authors have addressed all of my concerns, and I think that they have done an exemplary job of addressing the concerns raised by the other reviewers. The combination of data shown in this manuscript are compelling and interesting, deepening our understanding of excitation-transcription coupling.

We thank the reviewer for the very positive comments.

Reviewer #2 (Remarks to the Author):

The authors have fully addressed all comments and have provided a high quality paper which will be of significant interest to the field.

We thank the reviewer for the very positive comments.

Reviewer #3 (Remarks to the Author):

The authors adequately revised the manuscript. I totally understand that the first point (kinase vs carrier role) has been already addressed in their earlier paper.

We thank the reviewer for the positive comment and for acknowledging that we had already addressed the issue of kinase vs carrier role in our earlier *Cell* paper.